# A New Bioreactor to Promote Human Follicular Growth with or without Activin A in Transgender Men

Cynthia Jovet [1,2], Eloïse Fraison [1,2,3], Jacqueline Lornage [1,3,4], Nicolas Morel Journel [5], Antoine Gavoille [6,7], Laurent David [8], Alexandra Montembault [8], Cyrielle Fournier [2], Bruno Salle [1,3,4] and Elsa Labrune [1,2,3,*]

1   Hospices Civils de Lyon, Hôpital Mère Enfant, Service de Médecine de la Reproduction, 59 Boulevard Pinel, 69677 Bron, France
2   Faculté de Médecine Laennec, Université Claude Bernard, 7 Rue Guillaume Paradin, 69100 Lyon, France
3   INSERM Unité 1208, 18 Avenue Doyen Lépine, 69500 Bron, France
4   Faculté de Médecine Lyon Sud, Université Claude Bernard, 165 Chemin du Petit Revoyet, 69921 Oullins, France
5   Hospices Civils de Lyon, Service Urologie, 69310 Lyon, France
6   Campus La Doua, Université Claude Bernard Lyon 1, 43 Boulevard du 11 Novembre 1918, 69622 Villeurbanne, France
7   Hospices Civils de Lyon, Service de Biostatistique-Bioinformatique, 69500 Lyon, France
8   Ingénierie des Matériaux Polymères (IMP), CNRS UMR 5223, Université de Lyon, Université Claude Bernard Lyon1, Institut National des Sciences Appliquées de Lyon, Université Jean Monnet, CNRS, UMR 5223, 15 Boulevard A. Latarjet, 69622 Villeurbanne, France
*   Correspondence: elsa.labrune@chu-lyon.fr; Tel.: +33-(0)4-72-12-94-12

**Abstract:** The aim of the present study was to evaluate the effect of activin A on the activation of in vitro folliculogenesis of human ovarian tissues from transgender men with or without our new compartmented chitosan hydrogel microbioreactor ("three-dimensional (3D)-structure") enabling a three-dimensional tissue culture. Five fresh ovarian human tissues were cultured in vitro for 20 or 22 days in four groups with 100 ng/mL activin A or without activin A during the last six to eight days of culture, and within a 3D-structure or without the 3D-structure in standard conditions. Follicular density and quality were evaluated, and follicular diameters were measured. Estradiol secretion was quantified. Proliferation and apoptosis through immunostaining were also performed. The proportion of primordial follicles was significantly reduced, and the proportion of primary and secondary follicles was significantly increased in all four groups ($p < 0.001$). Tertiary follicles were observed in the four culture groups. Activin A supplementation did not significantly affect the follicular density, follicular quality, follicular growth, or estradiol secretion ($p > 0.05$). The 3D-structure increased the density of primary follicles and decreased the estradiol secretion ($p < 0.001$). Follicular proliferation was significantly lower in the 3D-structure group compared to the non-3D-structure group ($p = 0.008$). Regarding follicular apoptosis, it was significantly higher in the activin group compared to the non-activin group ($p = 0.006$). Activin A did not seem to play a key role in the in vitro folliculogenesis activation in our culture conditions. However, the results may indicate that the 3D-structure could be more physiological and could prevent a detrimental in vitro folliculogenesis flare-up.

**Keywords:** in vitro folliculogenesis; bioreactor; activin A; follicle growth; ovarian tissue culture





## 1. Introduction

Ovarian tissue cryopreservation and autograft is the only option available for pre-pubertal girls and patients who cannot delay their cancer treatment. However, in acute leukemia, for example, there is a risk of reintroducing malignant cells with the ovarian tissue transplantation [1]. Therefore, in vitro systems have been developed to support the growth and maturation of follicles, a fortiori oocytes, from immature to the final stage. The first live birth resulting from complete in vitro folliculogenesis was obtained in mice in

1996 by Eppig et al. [2] and was repeated in 2003 [3]. In large animals, the results were encouraging too; recently, the first positive pregnancy diagnosis in a goat from in vitro follicle culture was reported [4]. However, no team reported having obtained mature oocytes from primordial follicles [5]. In human, metaphase II oocytes have been obtained, due to a multi-step culture system using activin A, opening the possibility of in vitro fertilization as an option for women who cannot use the transplantation of their own ovarian tissue [6].

Several culture systems are currently used to support in vitro folliculogenesis of isolated follicles: Matrigel® (BD Bioscience, San Jose, CA, USA), gelatin structure, alginate drop, or synthetic extracellular matrix-like components. Most culture systems are currently based on the enzymatic or mechanical isolation of follicles [7–11]. An alternative approach is to work on ovarian fragments rather than on an isolated follicle, in order to maintain the architecture and intercellular interactions of the follicles. The use of activin A combined with a proper culture system was reported to activate the primordial follicles and to initiate follicle growth [12]. Activin A is a dimeric glycoprotein belonging to the transforming growth factor beta superfamily and its receptors are expressed in primordial, primary, and secondary follicles. It stimulates the granulosa cell proliferation, follicle, and oocyte growth and also accelerates oocyte maturation during culture of ovary cortex [13,14].

Our team is working on an innovative 3D-structure made from chitosan hydrogel with an inner compartment [15], which can maintain the integrity of the ovarian tissue [16]. The ovarian fragment is placed into a bioreactor chamber, which is sealed on each end. This encapsulated system enables the paracrine factors to be maintained within the ovarian tissue.

The aim of this study was, therefore, to evaluate the activation of in vitro folliculogenesis with or without activin A during the last six to eight days of culture, within or without our new 3D-structure of human ovarian tissue.

## 2. Methods

All study procedures adhered to the relevant ethical guidelines and we obtained the approval of the local ethics committee on 27 February 2018, project N° 19-138. A written consent form to participate, and for publication, was obtained from all patients.

### 2.1. Ovarian Tissue Collection

Fresh ovaries ($n$ = 5) from 5 transgender men were obtained with informed consent during their oophorectomy (Centre Hospitalier Lyon Sud, urology department). Mean ± standard deviation age of the patients was 25 ± 7 years. They all received a treatment by testosterone with a mean ± standard deviation duration of 25.3 ± 18.4 (4–43) months. Approval of this study was given by the local ethics committee. Ovarian tissue was transported to the laboratory in dissection medium (Minimum Essential Medium, Alpha 1X, Corning Life Sciences, New York, NY, USA) within 30 min at the temperature of 4–10 °C.

### 2.2. Bioreactor Preparation

The new 3D-structure was a tube-shaped hydrogel made from chitosan, license number FR1910681 [15]. Chitosan hydrogel bioreactors were prepared as described in 2008 by David et al. and Araizi et al. [17,18]. Chitosan was first purified as previously described by Montembault et al. [19], then chitosan hydrogels were prepared by the gelation of aqueous chitosan acetate solutions. Briefly, purified chitosan was dissolved in an acetic acid solution at a polymer concentration of 0.5%, and then filtered successively through 5, 3, 0.8, and 0.42 µm filters (Millipore, Molsheim, France). Filtered chitosan acetate was next precipitated with ammonia, washed with deionized water, and freeze-dried. Later, a chitosan solution at 2% *w/w* polymer concentration was obtained by the dispersion of neutralized chitosan lyophilizate in water plus acetic acid added in stoichiometric amounts sufficient to achieve protonation of –NH$_2$ groups of glucosamine residues. That solution was centrifuged in dispenser syringes (Nordson EFD, Westlake, OH, USA) and extruded through a 6-mm diameter tip using Performus I dispenser (Nordson EFD). The extrudate was neutralized

for 2 min in sodium hydroxide 1 M to coagulate the external part of the extrudate. This resulted in a tubular hydrogel that was stabilized in deionized water. The uncoagulated inner solution was removed by an airflow creating a hollow tube. These 4-mm diameter and 1-mm membrane 'bioreactors' ("3D-structure") were sterilized for 20 min at 121 °C in an autoclave and stored until use in deionized water at room temperature. The ovarian tissue fragments were placed into the lumen of the bioreactor tubes and the extremities were sealed with suture wire on each end (Figure S1).

### 2.3. Tissue Preparation and Ovarian Fragments Culture

Following transport, the ovarian tissue was transferred to a fresh dissection medium (Minimum Essential Medium, Alpha 1X, Corning Life Sciences, New York, NY, USA) and then cut lengthwise in two fragments. For each fragment, the medulla was removed using flat dissection scissors, thus obtaining two hemi cortexes. These hemi cortexes were dissected into ovarian fragments of ~ 3 × 3 × 1 mm.

For each ovary, one fresh fragment was fixed in 4% paraformaldehyde at room temperature during 24 h for histological evaluation (corresponding to the control at Day 0), whereas the remaining fragments were divided randomly in four culture groups (Figure 1):

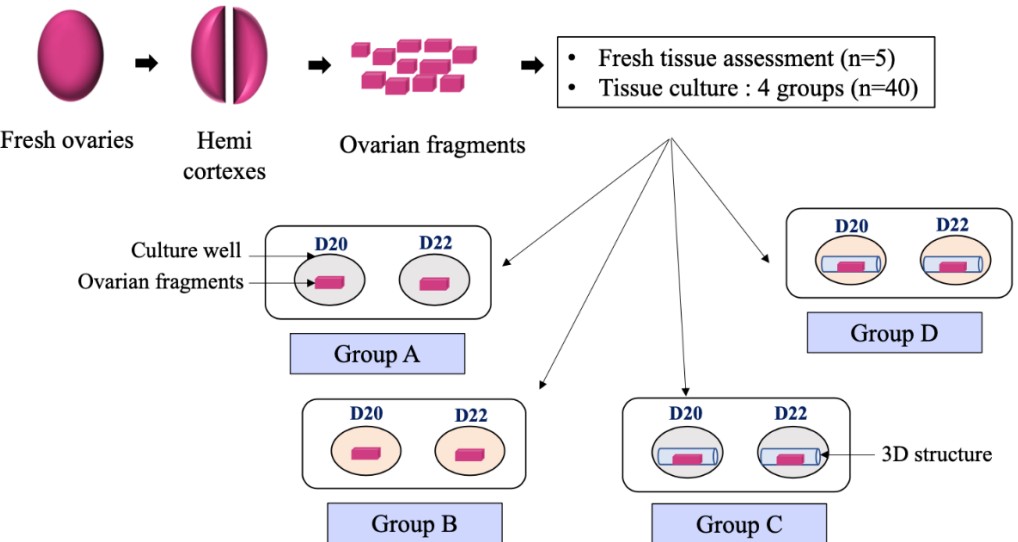

**Figure 1.** Design of the work.

The fresh ovary was cut lengthwise in two fragments. For each fragment, the medulla was removed, thus obtaining two hemi cortexes. These hemi cortexes were dissected into ovarian fragments. One fresh fragment was taken for fresh tissue assessment, whereas the remaining fragments were placed randomly in four culture groups for 20 (D20) or 22 days (D22):

- Group A (3D−/A−): Ovarian fragment placed in culture medium without activin A without 3D-structure
- Group B (3D−/A+): Ovarian fragment placed in culture medium with activin A but without 3D-structure
- Group C (3D+/A−): Ovarian fragment placed in culture medium without activin A but within a 3D-structure
- Group D (3D+/A+): Ovarian fragment placed in culture medium with activin A within 3D-structure.

Activin A (A4941-10UG, Sigma-Aldrich, Saint Louis, MO, USA) was used at the dose of 100 ng/mL, according to the literature. Ovarian fragments with any visible antral follicles under light microscopy were not included.

8 fragments (of ~ 3 × 3 × 1 mm) from each patient (*n* = 5) were cultured in A, B, C, and D groups (2 fragments of the same patient per group). Thus, a total of 40 fragments were incubated individually in cell culture plates containing 3 mL of culture MEM Alpha Medium (Corning LifeSciences, New York, NY, USA) supplemented with L-glutamine, 10% of serum substitute supplement (SSS; Irvine scientific, Santa Ana, CA, USA), 100 UI/mL of penicillin G, 100 μg/mL of streptomycin (HyClone, South Logan, OH, USA), 2.5 μg/mL of amphotericin B (PAN-Biotech, Bayern, Germany), 10 ng/mL of insulin, 10 ng/mL of transferrin, $10^{-3}$ μg/mL of selenium (PAN-Biotech), 25 mUI/mL of rFSH (Bemfola®, Gedeon Richter, Budapest, Hungary), 50 μg/mL of ascorbic acid (Sigma-Aldrich, St Quentin Fallavier, France), and 1 g/L of glucose (Sigma-Aldrich, St Quentin Fallavier, France).

Ovarian fragments assigned to groups A and B were placed in culture wells, whereas ovarian fragments assigned to groups C and D were placed into 3D-bioreactors before been placed in the culture wells.

Fragments were cultured in a two-stage procedure: 14 days of standard media, followed by 6 or 8 days of activin A supplementation (for group B and D). The total culture time was therefore 20 or 22 days. The fragments were initially cultured for 14 days at 37 °C in humidified air with 5% $CO_2$, with half the medium being changed every two days. This first stage of culture was set up because of the thickness of the ovarian fragments, to allow the atresia of the remaining antral follicles not visible under light microscopy. Previous studies in humans have tested an activin A supplementation duration of 2, 4, 7, or 8 days [12,14,20]. With better reported results for the longest supplementations dates, we assessed two different supplementation conditions: 6 and 8 days.

### 2.4. Histological Evaluation

Fresh and cultured ovarian tissues were fixed in paraformaldehyde 4% (Sigma-Aldrich, Saint Louis, MO, USA), dehydrated, included in paraffin, and cut into 3μm serial sections. Five sections were cut per ovarian fragment, equidistant from 60 μm each, except for a 200 μm gap between the 2nd and the 3rd section. The five sections per ovarian fragment were observed by light microscopy MOTIC Panthera (PantheraCC®, MoticEurope, Barcelone, Spain) after being stained with hematoxylin and eosin (Sigma Alrich, Saint Louis, MO, USA). A total of five fragments of fresh cortical tissue and 40 fragments of cultured tissue were fixed in paraformaldehyde 4% and processed for histological evaluation as described above. A total of 3519 follicles were examined under the light microscope (Table 1).

### 2.5. Follicular Density

The follicular density was compared between the fresh group and groups A, B, C, and D after 20 or 22 days of culture. Follicles were classified according to their stage of development based on A. Gougeon's [21] classification as follows:

Primordial stage: oocyte surrounded by flattened granulosa cells,
Intermediate stage: oocyte surrounded by flattened and cuboidal granulosa cells,
Primary stage: oocyte surrounded only by cuboidal granulosa cells,
Secondary stage: oocyte surrounded by 2 or more layers of cuboidal granulosa cells,
Tertiary stage: antrum formation.

Follicular density of a tissue is defined as the number of follicles divided by the tissue surface. The integrated measuring tool in the ImageJ software (1.52 version) was used to measure fragments' dimensions for follicle density. The follicles were photographed under the microscope and the photos were opened in the image-processing software. The 2 points for calculating the diameter were determined manually according to the organization of the granulosa cells. The measurement tool integrated in the software determined the value of the follicular diameter between these 2 points.

**Table 1.** Total number of follicles counted for primordial, intermediate, primary, secondary, and tertiary stage in fresh and after culture.

| Group | | | Total | Primordial Stage | | Intermediate Stage | | Primary Stage | | Secondary Stage | | Tertiary Stage | |
|---|---|---|---|---|---|---|---|---|---|---|---|---|---|
| | | | | Intact | Altered | Intact | Altered | Intact | Altered | Intact | Altered | Intact | Altered |
| Fresh | | Total number of follicles | **637** | 22 | 375 | 5 | 61 | 27 | 102 | 17 | 27 | 0 | 1 |
| | | Follicular stage by group (%) | | 3.5 | 58.9 | 0.8 | 9.6 | 4.2 | 16.0 | 2.7 | 4.2 | 0 | 0.2 |
| D20 | A | Total number of follicles | 265 | 3 | 38 | 4 | 59 | 25 | 56 | 61 | 18 | 1 | 0 |
| | | Follicular stage by group (%) | | 1.1 | 14.3 | 1.5 | 22.2 | 9.4 | 21.1 | 23.0 | 6.8 | 0.4 | 0 |
| | B | Total number of follicles | 301 | 16 | 118 | 10 | 41 | 12 | 17 | 37 | 46 | 1 | 3 |
| | | Follicular stage by group (%) | | 5.3 | 39.2 | 3.3 | 13.6 | 4.0 | 5.6 | 12.3 | 15.2 | 0.3 | 1.0 |
| | C | Total number of follicles | 423 | 2 | 103 | 1 | 172 | 24 | 66 | 28 | 23 | 2 | 2 |
| | | Follicular stage by group (%) | | 0.5 | 24.3 | 0.2 | 40.7 | 5.7 | 15.6 | 6.6 | 5.4 | 0.5 | 0.5 |
| | D | Total number of follicles | 315 | 11 | 9 | 33 | 36 | 85 | 56 | 47 | 34 | 2 | 2 |
| | | Follicular stage by group (%) | | 3.5 | 2.9 | 10.5 | 11.4 | 27.0 | 17.8 | 14.9 | 10.8 | 0.6 | 0.6 |
| D22 | A | Total number of follicles | 535 | 13 | 66 | 35 | 107 | 123 | 89 | 84 | 17 | 1 | 0 |
| | | Follicular stage by group (%) | | 2.4 | 12.3 | 6.5 | 20.0 | 23.0 | 16.6 | 15.7 | 3.2 | 0.2 | 0 |
| | B | Total number of follicles | 347 | 5 | 85 | 14 | 109 | 50 | 29 | 33 | 20 | 1 | 1 |
| | | Follicular stage by group (%) | | 1.4 | 24.5 | 4.0 | 31.4 | 14.4 | 8.4 | 9.5 | 5.8 | 0.3 | 0.3 |
| | C | Total number of follicles | 229 | 0 | 2 | 10 | 14 | 71 | 59 | 52 | 20 | 0 | 1 |
| | | Follicular stage by group (%) | | 0.0 | 0.9 | 4.4 | 6.1 | 31.0 | 25.8 | 22.7 | 8.7 | 0 | 0.4 |
| | D | Total number of follicles | 467 | 5 | 59 | 21 | 42 | 150 | 125 | 49 | 13 | 1 | 2 |
| | | Follicular stage by group (%) | | 1.1 | 12.6 | 4.5 | 9.0 | 32.1 | 26.8 | 10.5 | 2.8 | 0.2 | 0.4 |

Fresh: One fresh fragment per patient was evaluated before tissue culture; D20: The fragments were cultured for 20 days at 37 °C in humidified air with 5% C02; D22: The fragments were cultured for 22 days at 37 °C in humidified air with 5% C02; Group A (3D−/A−): Ovarian fragment placed in culture medium without activin A without 3D-structure; Group B (3D−/A+): Ovarian fragment placed in culture medium with activin A but without 3D-structure; Group C (3D+/A−): Ovarian fragment placed in culture medium without activin A but within a 3D-structure; Group D (3D+/A+): Ovarian fragment placed in culture medium with activin A within 3D-structure.

## 2.6. Follicular Growth

Secondary stage follicles, i.e., oocyte surrounded by 2 or more layers of cuboidal granulosa cells, were measured with the integrated measuring tool in the Image J software (version 1.53o). The mean follicle diameters were compared between the fresh group and groups A, B, C, and D after 20 or 22 days of culture.

## 2.7. Follicular Quality and Morphology

The proportions of intact and atretic follicles were compared between the groups. The follicles were classified as follows:

Intact follicle: intact follicle with a complete layer of granulosa cells and no alteration criteria.

Altered follicle: empty follicle, cytoplasmic detachment, oocyte degeneration, incomplete layer of granulosa cells (Figure 2).

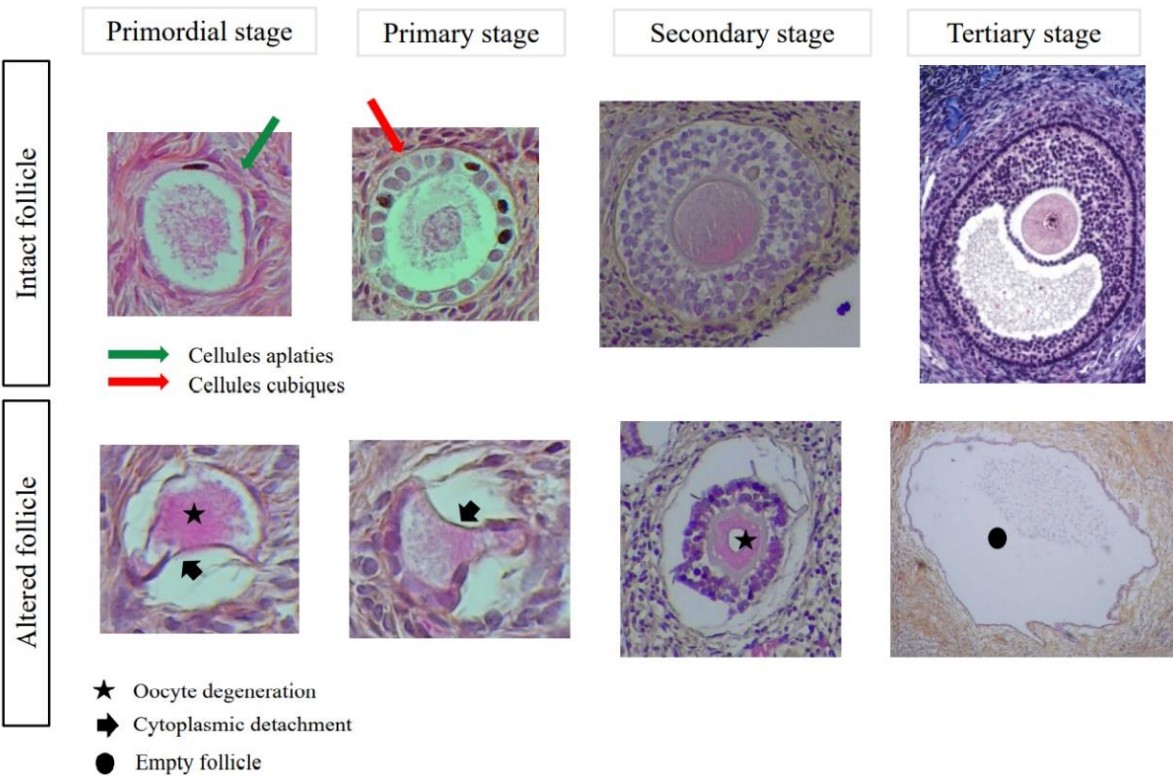

**Figure 2.** Classification (from Gougeon) according to human follicle morphology: intact tertiary follicle [21] altered tertiary follicle (personal pictures) × 100; all other follicles (personal picture) × 400.

The follicular quality of the fresh tissue was impaired compared to the literature [20]. Therefore, the follicular qualities between our 4 culture groups were compared to each other, not to the fresh state.

## 2.8. Detection of Ki67 and Cleaved Caspase 3 Immunoreactivity in Cultured Follicles

To evaluate follicular proliferation and apoptosis, Ki67 (9027; Cell Signaling Technology, Danvers, MA, USA) and Cleaved Caspase 3 (9664S; Cell Signaling Technology) immunoreactivity were used. Slides in every culture group previously described were chosen. Only the slides with the greatest numbers of follicles seen in histology were selected to perform immunostaining. Every slide contained two tissue sections; the upper one was used for Ki67 and the lower one for Cleaved Caspase 3. Each series had a negative and a positive control slide.

Ovarian tissue sections were dewaxed in methylcyclohexan (Sigma-Aldrich) and absolute ethanol (Sigma-Aldrich), and then rinsed with water. The unmasking was carried out by an immersion of the slides in sodium citrate buffer (pH = 6 (Merck, Darmstadt, Germany)) heated in a water bath for 45 min at 98 °C. The detection of the antigen was realized due to a kit based on peroxidase using 3, 3′-Diaminobenzidine (DAB) used as chromogen (EnVisionTM GI2 Doublestain System®; Dako, Glostrup, Denmark). Tissue sections were incubated with Dual Endogenous Enzyme Block to block endogenous peroxidase and were then washed in Phosphate Buffered Saline (PBS). The sections were probed with the primary antibody for one hour at room temperature, washed with PBS, and then incubated with the Polymer/HRP reagent. DAB was finally applied for 15 min after a last washing with PBS. Counterstaining was achieved using standard hematoxylin staining (Millipore). Samples were mounted using an aqueous mounting medium (Dako Faramount®; Dako). The two primary antibodies used were a monoclonal rabbit anti-Ki 67 antibody (9027; Cell Signaling Technology, Danvers, MA, USA), which is a specific marker of cell proliferation with a dilution of 1/500, and a monoclonal rabbit anti-cleaved caspase-3 antibody (9664S; Cell Signaling Technology), which is a specific marker of apoptotic cells with a dilution of 1/200. Two readers independently observed the slides using light microscopy (Panthera CC®; Motic Europe, Barcelona, Spain).

Tissue proliferation and apoptosis were expressed by a cell density. Marked cells with DAB chromogen were counted per tissue section surface. Follicle proliferation was assessed with a percentage of positive follicles over the total number of follicles. A follicle was considered as positive when at least one granulosa cell was stained. Regarding apoptosis, the follicle was considered in apoptosis if more than 50% of the cells were brown. In both cases, the oocyte was considered as a major cell and when positive, the follicle was judged 100% in proliferation or apoptosis.

All the slides were read by two independent readers for histological evaluation, Ki67 and Cleaved Caspase 3 detection.

### 2.9. Detection of Estradiol in Culture Medium

Concentrations of estradiol in culture media were determined using a Chemiluminescent Microparticle Immunoassay CMIA (Abbott) at day 6, day 20, and day 22. The functional sensitivity of the estradiol assay was <25 pg/mL.

### 2.10. Statistical Analyses

Continuous variables are described with the mean, standard deviation, and categorical variables with the count and percentage. To manage the non-independence of the data, the effects of activin A and 3D-structure were analyzed using multivariate models considering the patients and the culture conditions (culture time and level). Follicular density, the proportion of marked follicles and the density of cells marked by immunostaining (Ki67 and Cleaved Caspase 3) were analyzed with a zero-inflated binomial negative model, suitable for counting over-dispersed data. The follicular diameter and estradiol level were analyzed with a linear regression model after logarithmic transformation. Due to non-homoscedastic residual variance, variance was modeled by considering the patient for the follicular diameter and the expected value for the estradiol level. A *p*-value less than 0.05 was considered significant. The analyses were performed using the R software, version 3.6.1 with the "pscl" library.

## 3. Results

### 3.1. Follicular Density

A total of 637 follicles in the fresh group and 1304 and 1578 follicles in the culture of days 20 and 22, respectively, have been counted and classified in function of their quality (intact versus altered), their stage (primordial, primary, secondary, and tertiary), and the culture conditions (with or without activin, with or without 3D structure) (Table 1).

A decrease in follicular density was observed between the fresh and the cultured fragments (3.8 ± 2.0 follicles/mm$^2$ vs. 1.7 ± 1.6 and 2.1 ± 1.8 follicles/mm$^2$ (at days 20 and 22), $p < 0.001$). No significant difference was observed between the group with activin compared to the group without (1.9 ± 1.6 follicles/mm$^2$ vs. 1.9 ± 1.8 follicles/mm$^2$, $p > 0.05$). No significant difference was observed between the group with 3D-structure compared to the group without (1.9 ± 1.7 follicles/mm$^2$ vs. 1.9 ± 1.7 follicles/mm$^2$, $p > 0.05$).

The proportion of primordial follicles was significantly reduced through culture but not affected by activin A supplementation or 3D-structure, $p < 0.001$ (Table 1 and Figure 3A).

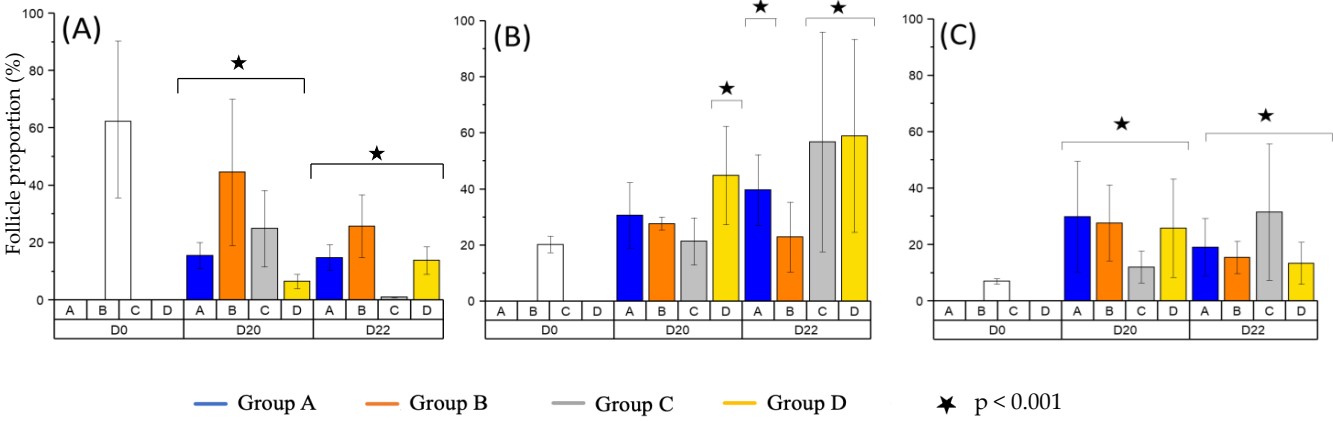

**Figure 3.** (**A**) Proportion of primordial follicles in the control fragment (D0) versus the cultured fragments (D20 and D22). (**B**) Proportion of primary follicles in the control fragment (D0) versus the cultured fragments (D20 and D22) and (**C**) Proportion of secondary follicles in the control fragment (D0) versus the cultured fragments (D20 and D22).

The proportion of primary follicles was significantly increased through culture in group D at D20 and in groups A, B, and D at D22 (Table 1 and Figure 3B).

The proportion of secondary follicles was significantly increased through culture, $p < 0.001$ (Figure 3C).

Tertiary follicles were observed in the four cultured groups (one tertiary follicle in fresh group versus 20 tertiary follicles in cultured groups).

After comparing the groups with (groups B + D) and without activin A (groups A + C), activin A supplementation did not significantly affect the follicular distribution between primordial, primary, and secondary follicles, $p > 0.05$. However, after comparing the groups with (groups C + D) and without 3D-structure (groups A + B), the 3D-structure significantly increased the density of primary follicles (RR = 1.39, IC95%: 1.14–1.70, $p < 0.001$) (Table 1 and Figure 3).

*3.2. Follicular Growth*

Ovarian cortex fragments were cultured for 20 or 22 days in four culture groups. 626 secondary follicles were measured, whatever their quality. No significant difference of follicle diameter was observed between day 0 (70.0 ± 19.2 μm) and the four groups at day 20 (63.6 ± 18.9 μm) or the four groups at day 22 (70.4 ± 28.4 μm).

However, at day 20 and day 22, the diameters of secondary follicles cultured without a 3D-structure were greater (69.6 ± 21.2 μm) than those cultured within a three-dimensional structure (63.5 ± 27.2 μm) ($p < 0.001$). Activin A supplementation did not significantly influence the mean diameter of follicles (65.7 ± 29.1 μm with activin A vs. 68.0 ± 18.0 μm without, $p > 0.05$).

### 3.3. Follicular Quality and Morphology

The mean fraction of intact follicles in the cultured fragments was observed (Table 1, Figures S2 and S3). Follicular quality indicated no difference between the groups, and no difference in function of the follicular stages, $p > 0.05$. The presence of activin A or 3D-structure did not significantly affect follicular quality, $p > 0.05$ (Table 2).

**Table 2.** Fraction of intact follicles, all stages combined, counted in the cortex fragments after 20 days and 22 days of culture.

| Group | D20 | | D22 | |
| --- | --- | --- | --- | --- |
| | % Intact Follicles (Mean ± SD) | *n* | % Intact Follicles (Mean ± SD) | *n* |
| A (3D−/A−) | 42.7 ± 34.7 | 265 | 54.3 ± 33.6 | 535 |
| B (3D−/A+) | 27.7 ± 16.4 | 301 | 35.7 ± 33.0 | 347 |
| C (3D+/A−) | 17.5 ± 29.1 | 423 | 62.9 ± 32.7 | 229 |
| D (3D+/A+) | 54.5 ± 5.0 | 315 | 54.1 ± 31.4 | 467 |

*n*: total number of follicles counted.

### 3.4. Estradiol Secretion

Media from ovarian fragments cultured individually were analyzed for estradiol content by chemiluminescent microparticle immunoassay (CMIA). The estradiol levels are described in Table 3.

**Table 3.** Estradiol levels (mean ± SD; pg/mL) in culture media of cultured ovarian fragments (D: days of culture).

| Group | D6 | | D20 | | D22 | |
| --- | --- | --- | --- | --- | --- | --- |
| A (3D−/A−) | 120.0 | ±55.43 | 5151.0 | ±3463.23 | 3323.6 | ±3780.42 |
| B (3D−/A+) | 146.0 | ±10.39 | 2227.2 | ±1736.04 | 1738.6 | ±1002.98 |
| C (3D+/A−) | 237.3 | ±162.24 | 446.8 | ±326.12 | 1130.2 | ±683.07 |
| D (3D+/A+) | 680.3 | ±551.95 | 854.2 | ±990.17 | 1965.0 | ±1650 |

A significant increase in estradiol secretion was observed in cultured ovarian fragments at days 20 and 22 compared with the day 6 group ($p < 0.001$) (Figure 4). Estradiol levels were estimated to be 1795.6 ± 2680.7 pg/mL in the activin A-supplemented groups versus 1287.5 ± 1170.5 pg/mL in the non-activin A groups. There was no statistically significant difference between the groups (RR = 1.33, CI95%: 0.91–1.94, $p > 0.05$). The presence of activin A did not significantly alter the estradiol level of the culture medium. Significant estradiol secretion was observed in ovarian fragments cultured without 3D-structure. Estradiol levels were measured at 802.8 ± 904.3 pg/mL in the presence of the 3D-structure and 2280.3 ± 2597.9 pg/mL without. A decrease of 2.5 in estradiol levels was observed in the presence of the 3D-structure ($p < 0.001$).

### 3.5. Detection of Ki67 Antigen

Sixty slides with 60 tissue sections were evaluated for Ki67, equally distributed in each four groups. Tissue and follicle proliferation were confirmed by the detection of the proliferating nuclear antigen Ki67 on several growing follicles and tissue cells from the control slides (Figure S4). The mean cellular density of Ki67-positive stromal cells between the ovarian tissue was significantly less important in the group with 3D-structure than in the group without (7.1 ± 12.3 cells/mm$^3$ vs 14.5 ± 21.7 cells/mm$^3$, $p = 0.008$). The difference between the group with activin A compared to the group without was not significant (7.4 ± 12.1 cells/mm$^3$ vs 14.1 ± 22.0 cells/mm$^3$, $p = 0.544$).

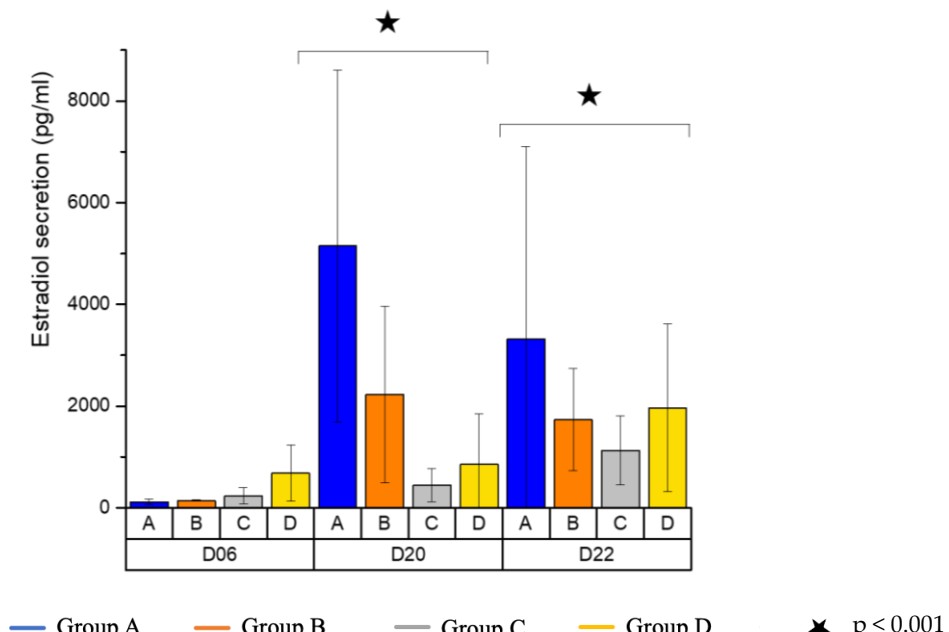

**Figure 4.** Estradiol secretion after 6, 20, and 22 days of in vitro culture. The ovarian fragments analyzed are different in the D20 culture versus the D22 culture, although obtained from the same ovary.

The difference regarding the mean number of positive follicles for proliferation was not significant within the group with the 3D-structure compared to the group without ($3.2 \pm 8.6\%$ vs. $14.9 \pm 26.6\%$, $p = 0.081$) and in the group with activin A compared to the group without ($5 \pm 17.1\%$ vs. $12.8 \pm 22.5\%$, $p = 0.108$) (Table 4).

**Table 4.** Number of counted follicles in each condition (3D−: culture without 3D structure; 3D+: culture within 3D structure; A−: culture without Activin A; A+: culture with Activin A).

|  |  | Number of Counted Follicles | Number of Positive Follicles (Ki67+) |
|---|---|---|---|
| Ki67 | 3D− | 379 | 33 |
|  | 3D+ | 384 | 12 |
|  | A− | 413 | 53 |
|  | A+ | 350 | 17 |

*3.6. Detection of Cleaved Caspase 3 Antigen*

The same slides were used to evaluate the presence of Cleaved Caspase 3. Tissue and follicle apoptosis were confirmed by the detection of the apoptosis nuclear antigen Cleaved Caspase 3 on several wasted follicles and tissue cells from the control slides.

The difference in cellular density of caspase 3-positive stromal cells within the ovarian tissue was not significant between the group with the 3D-structure compared to the group without ($5.1 \pm 5.3$ cells/mm$^3$ vs. $9.6 \pm 14.8$ cells/mm$^3$, $p = 0.057$). However, it was significantly more important in the group with activin A compared to the group without ($10.4 \pm 14$ cells/mm$^3$ vs. $3.5 \pm 2.9$ cells/mm$^3$, $p = 0.006$).

The difference regarding the mean number of positive follicles for apoptosis was not significant between the group with the 3D-structure compared to the group without ($10.1 \pm 14.8\%$ vs. $14.5 \pm 24.3\%$, $p = 0.318$) and in the group with activin A compared to the group without ($13.8 \pm 22.7\%$ vs. $10.3 \pm 15.7\%$, $p = 0.859$) (Table 5).

**Table 5.** Number of counted follicles in each condition (3D−: culture without 3D structure; 3D+: culture within 3D structure; A−: culture without activin A; A+: culture with activin A).

|         |     | Number of Counted Follicles | Number of Positive Follicles |
|---------|-----|-----------------------------|------------------------------|
| Caspase | 3D− | 379                         | 55                           |
|         | 3D+ | 384                         | 39                           |
|         | A−  | 413                         | 42                           |
|         | A+  | 350                         | 48                           |

## 4. Discussion

The objective of this study was to evaluate a new 3D-structure and the impact of activin A during the last part of culture on the activation of folliculogenesis in human ovaries' ovarian cortex. We had previously demonstrated the survival of more than three months of human ovarian tissue in this microbioreactor [16]. In this study, we found a decrease in follicular density (all follicular stages combined) between the fresh and the cultured fragments. Our results are in accordance with the literature, and have already been described in the study of Ding et al. [20]. It may be partly explained by an alteration of the tissue after the surgery. Even though the time and transit of the ovaries were optimized, the interruption of the vascularization is probably responsible for some tissue damage. It should also be explained by the fact that culture conditions are different from physiological conditions. Simultaneously, spontaneous folliculogenesis was observed in all four groups regardless of the presence of the 3D-structure or Activin A. The repressive, mechanical, and compressive action of ovarian tissue on folliculogenesis in the human body through inhibitory factors has already been described [14,22–24]. Moreover, it is already well known that ovarian fragmentation, by disrupting the Hippo signaling pathway, induces follicular growth [25]. In all four groups, we found a decrease in primordial follicles and an increase in primary and secondary follicles. These results are aligned with other studies on the subject [6,14,20]. Nonetheless, it must be underlined that we found a more important density of primary follicles with the 3D chitosan hydrogel structure than without, which has never been described before. Moreover, we found tertiary follicles in all groups. Many authors found a positive impact of activin A for the development of tertiary follicles [6,14,20]. We did not find this effect in our study.

Regarding the mean follicular diameter, it was significantly smaller in the 3D-structure groups than in the groups without 3D microbioreactor structure. This effect was already described in the Higuchi et al. study with the use of Matrigel® [26]. Two hypotheses may explain this difference. The first one is the necrosis of the follicles creating a swelling and a disruption of granulosa cells leading to an increase in their size before the scattering of the cytoplasmic and nuclear contents. The second one could be the disjunction of the tissue cultivated without any structure. The expression of connexin 43 was already described in two studies [6,27] showing an increase in this protein in the group with activin A. Again, we did not find an impact of activin A on follicular diameters.

Regarding follicular quality, no significant difference was found between the four groups. Ding et al. and Telfer et al. did not find a positive impact of activin A on follicular quality, whereas it was described in mice and sheep [14,20,28,29]. It must be mentioned that even though quality and follicular morphology were evaluated by two readers, human assessment always remains subjective.

Concerning the assessment of the steroidogenesis function of the ovary, we measured the estradiol level in culture media. We observed an increase in the estradiol level in all groups evidencing the differentiation and functionality of the granulosa cells. This is in agreement with previous studies [14]. More importantly, we found a higher level of steroids in both groups without any structure. Here again, two hypotheses are raised to explain that estradiol level is less important in the 3D-structure. The first one could be the releasing of estradiol after cellular lysis from the cortex fragments cultured without

structure. The second one could be a detrimental uproar of the folliculogenesis with an increase in estradiol synthesis, in absence of the 3D bioreactor.

Regarding proliferation and apoptosis, we found more Ki67 and Cleaved Caspase 3-stained cells in both groups without 3D-structure. This supports our hypothesis of a non-physiological (too fast) folliculogenesis activation without 3D cultivation structure. Indeed, more apoptosis is found where more proliferation is present. Regarding activin A, we found the same proliferation and apoptosis pattern of the follicles in the groups with or without activin A. On the contrary, some authors found more proliferation with activin A [6], but apoptosis was not studied as technically as we studied it. Here, again, the results on activin A remains unclear.

Some limitations must be underlined in our study. The first one is the potential presence of secondary follicles at the beginning of the culture, which may have evolved into antral follicles. Those follicles could have been counted as new ones, despite the possibility that they were already there. We tried to remove them when they were at the antral stage. However, it is very likely that some of them at the secondary stage remained. Thus, the small antral follicles could produce AMH and could have inhibited the recruitment of primordial follicles. In the study of Mc Laughlin et al., the authors removed the follicles with a diameter superior to 40 μm [6]. Nonetheless, as in Ding et al., Higuchi et al., Cossigny et al., and da Silva et al., we always compared our cultured tissue to the fresh one [20,26,28,29].

Secondly, concerning our model, human ovaries were removed from patients who had testosterone before the surgery for a duration varying from four months to three years. This could have had an impact on the ovarian environment. Focusing on mice, it has been reported in the literature that testosterone treatment does not alter the quantity of follicle within the ovary and its effects are reversible [30,31]. As for oocyte and follicle quality, the follicles were able to grow like follicles not exposed to testosterone and allow oocyte maturation. The oocytes resulted in the formation of cleaved embryos [32]. In a recent oral conference, E Telfer obtained less follicles and a slower follicular growth in tissue exposed to male hormones than in non-exposed tissue (IVF worldwide, 2020). Moreover, another main limitation was the heterogeneity of the ovarian tissue concerning the ovarian reserve: a tissue with more follicles at baseline may be easier to activate than another tissue with a poor follicular reserve.

To finish, only five tissues were used. It could appear to be a small number of samples. Nevertheless, compared to other articles on in vitro folliculogenesis, it is already substantial. As a matter of fact, we studied at 3519 follicles, whereas in other studies, the numbers of follicles studied varied between 74 [14] and 1137 [20].

## 5. Conclusions

To conclude, our study found a spontaneous in vitro folliculogenesis activation after culture in all culture conditions. We did not find a real benefit of using activin A, which is quite different from other works on this subject. Notwithstanding, our study suggests that our 3D-hydrogel microbioreactor structure may prevent deleterious, excessive, and hasty activation of folliculogenesis. Activin A did not have any impact on follicular growth during the culture (up to 22 days) in the ovarian cortex fragment. Our new bioreactor seemed to allow a more physiological in vitro folliculogenesis activation. In vivo folliculogenesis is a really long process; creating a good follicle with a good oocyte needs time. To date, only two teams have succeeded in producing a mature follicle in humans [12,33]. We envision that this research will offer viable and better options for fertility preservation. Complete in vitro folliculogenesis was performed in mice and resulted in live birth [2,3]. No birth has been described in mammals and humans yet [5]. The results are promising but far from producing several good quality oocytes. Pregnancies and live births in mammals are essential to validate the current in vitro folliculogenesis techniques.

**Supplementary Materials:** The following supporting information can be downloaded at: https://www.mdpi.com/article/10.3390/reprodmed4010003/s1, Figure S1: Chitosan bioreactor; Figure S2: Secondary follicles after 22 days of culture; Figure S3: Tertiary follicles after 22 days of culture; Figure S4: Ki67 immunostaining.

**Author Contributions:** E.L. designed the study; N.M.J. performed the tissue extraction; C.J., E.F., and C.F. performed all the preparations of the tissue; A.G. made the statistics; C.J., E.F., and E.L. analyzed and interpreted the results; L.D., E.L., and A.M. designed the bioreactor; C.J. and E.F. wrote the manuscript and E.L., J.L. and B.S. corrected it. All authors have read and agreed to the published version of the manuscript.

**Funding:** This work was supported by the Agence de la Biomédecine under grant (Project number: R18122CC).

**Institutional Review Board Statement:** All study procedures met the relevant ethical guidelines and obtained the approval of the local ethics committee on 27 February 2018, project N° 19-138. A written consent form to participate and for publication was obtained from all patients.

**Informed Consent Statement:** Informed consent was obtained from all subjects involved in the study.

**Data Availability Statement:** The data that support the findings of this study are available from the corresponding author upon reasonable request.

**Conflicts of Interest:** The authors declare no conflict of interest.

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
