# Peer review of "A New Bioreactor to Promote Human Follicular Growth with or without Activin A in Transgender Men"

_2673-3897, doi:10.3390/reprodmed4010003_

Round 1

Reviewer 1 Report

The paper by Cynthia JOVET et al presents an interesting topic of generation of ‘artificial ovary’. As such, it is of a great importance to the field. Among assisted reproduction techniques, molecular scaffolds are used to reproduce a three-dimensional ovarian environment to improve the efficiency of in vitro follicle culture techniques. The work presented here focuses on the comparison of the in vitro culture of human cortex pieces in a tube-shaped hydrogel made from chitosan and a basic system without this scaffold. In addition, the potential action of Activin A during the last part of this culture is studied.

Unfortunately, in current form it is difficult to follow.

The experiments are rather superficially analysed using limited amount of techniques: it is mainly based on follicle morphology and quality, follicular density (at primordial, primary and secondary stage) and diameter of only secondary follicles.

The number of ovarian cortical samples (5 ovaries from 5 patients?) and ovarian fragments per ovary (8 in culture and a control one) seems acceptable. Given the heterogeneity of the tissue in the same ovary, a second control fragment would be better.

But we have no information about the total counts of follicles in each stage at D0, D20 or D22 except a global count of 3519 (line 166) in the method paragraph or 3499 follicles in the discussion (line 419) and also nothing about the healthy tertiary follicles (number, proportion and diameters) at D20 or D22. Their presence is only specified in one sentence line 273 and no proof, like a photomicrograph in situ in the cultured cortex piece has been given.

Furthermore, the status of the follicles in the cultured cortex was not tested by PCR: genes expressions would be interesting to analysed: Activin Receptor, AMH/AMHR, KIT/KITL, CYP19A1, CYP17A1 follistatin, FSH receptor, and activin bA and bB subunit … for example.

AMH and progesterone secretion in the medium would be also interesting to measure.

Photomicrographs of the bioreactor in place and ovarian cortical tissue before and at the end of culture must be submitted with at least a picture of healthy antral follicle at the end of the culture. Likewise, pictures of the immunostaining are lacking.

The analyses that are presented in this report are not enough to make the final conclusions and there is no proof of oocyte functionality with at least meiotic competence after a maturation step for example. That makes me think that any of the protocols tested are actually producing functional oocytes. Nevertheless, this new device of cortex culture is always interesting and contribute to a better understanding of the in vitro folliculogenesis process. However questions remain: Given the complexity of making this bioreactor, is this protocol really an advantage? How transferable is the technology?

The introduction must be expanded by adding and discussing published data on

- Activin A and its receptor with their expression according to the stages and ovarian location.

- ovarian cortex culture in different species

Below are more details:

               Abstract

Line 21-22: precise that Activin A is provided only during the 6-8 last days of culture and not all along the culture.

Line 25 ‘significantly increased in all four groups’ is not exact (see results section).

Line 26 ‘Antral cavities were observed in the four culture groups’ no evidence in results.

Line 31 (p=0.006)

  1. Introduction

Line 45: ‘who cannot use the transplantation of their own ovarian tissue’

Line 46: The reference is McLaughlin 2018 and not 2010

Describe first the most relevant published data about ovarian cortex culture then about individual isolated follicles culture systems while specifying the concerned species and distinguishing between review and research articles.

Line 47: to support in vitro folliculogenesis of isolated follicles

Line 55 : The use of Activin A combined with a proper culture system is reported to activate the primordial follicles and to initiate follicle growth during culture of ovary cortex + article reference

Line 59 : Bloise = Review / Telfer = individual follicle culture / Ding = cortex culture

Line 60 : It is a new paragraph: could you develop your concept of bioreactors with article references (perhaps for example Sacco et all, 2018 or other ones?)

Line 65 : with or without Activin A during the 6-8 last days of culture

Line 66 and 67 is a repeat of line 64-65. Choose!

  1. Methods

2.1. Ovarian tissue collection

Line 73 : have you collected 5 ovaries from 5 patients?

Line 76: give the range of months

2.2. Bioreactor preparation

I do not know the technique of ‘3D chitosan scaffold’ for the support of ovarian follicles growth in cortex so I cannot judge this part of the manuscript. The reference articles indicated in the manuscript correspond to a patent and do not help my level of understanding of this technical approach. This tissue engineering approach has been already validated in 2008 by the same research team.

2.3. Tissue preparation and ovarian fragments culture

Line 109: for each ovary, One fresh fragment was taken and fixed in 4% paraformaldehyde at room temperature during 24h for histological evaluation (corresponding to the control at Day0)

In the schema: Fresh tissue assessment n=5       Tissue culture : 4 groups n=40

Lines 121-128 are the repetition of lines 116-119

Line 154: precise that Activin A was tested on individual follicles.

2.5. Follicular density

Line 170 : A Gougeon article reference?

Line 177-178: explain the process of measure, rephrase

2.7. Follicular quality and morphology

Line 185: Precise if it is secondary or tertiary follicles

2.10. Statistical analyzes

Precise if you used SD or SEM in the data

3.      Results :

3.1. Follicular density

In freshly isolated uncultured tissue, the sum of the % of primordial, primary and secondary follicles is 89.5%. The remaining 10.5% correspond to the intermediate or tertiary follicles?

Indicate in your text the Figure 3

The number of follicles counted for each stage is of great importance, precise it.

For group B the data are different between the text and the figure 3:
Line 256: 44.5% in the text and ~40% in the figure 3
Line 263: 27.6% in the text and ~10% in the figure 3

Line 273: better placed in the 3.2 paragraph?

Line 275 : remove intermediate ?

Line 278-280: it is not the results but the description of the figure:
Rephrase like ‘proportion of primordial, primary and secondary follicles in the control ovarian fragments (D0) versus ….’ given the heterogeneity of the tissue in the same ovary

It would be easier to write Group A (3D-/A-) Group B (3D-/A+) etc…

Line 279: ‘The proportion of primary follicles was significantly increased by culturing (B)’, it is not true for the primary follicles in groups A, B and C at D20 and group B at D22

3.2. Follicular growth

Line 283: ovarian cortex fragments in place of follicles

Lines 284-285: Specify the number of follicles

Line 283-285: Precise: No significant difference
Precise: secondary follicles

3.3. Follicular quality and morphology

A table would be more readable

Specify the follicular stages concerned. Is there differences in function of the follicular stages?

What is the observation in the control fragment?

Line 315-317: It is a legend and not a summary of the results

Specify in the legend that the ovarian fragments analysed are different in the D20 culture versus D22, although obtained from the same ovary.

3.5. Detection of Ki67 antigen

Line 325-327: if there is a difference between the group with 3D-structure and the group without, it would be interesting to distinguish these 2 groups in the analysis of the effect of Activin.

The number of follicles counted in each conditions must be specify.

A photograph would be needed to illustrate.

3.6. Detection of Cleaved Caspase 3 antigen

Same remarks as for the previous paragraph

4. Discussion

Line 347: specify the presence or not of activin during the last part of culture

Line 348 : all follicular stages combined

Lines 354: there is also a mechanical and compressive action of the ovarian tissue.

The results must be better discuss and compare with the other culture of ovarian cortex.

The impact of testosterone on the ovary must be more explain with the researches already published in animals (mice…) or sportive women.

Lines 421-431 must be in conclusion

5. Conclusion:

Too short and inexact

Line 433 replace beginning by end of the culture. Add in the ovarian cortex fragments.

Open your conclusion with the lines 421-431 and add sentences in order to think further on the use of this technology already successfully achieved in the mice (references in nature) but very far of the success in mammals and especially in humans

Author Response

Manuscript ID: reprodmed-2067408

Title: A new bioreactor to promote follicular growth with or without Activin A

Answers to Reviewer 1

The authors thank the reviewer for his/her attentive examination of the manuscript.

The paper by Cynthia JOVET et al presents an interesting topic of generation of ‘artificial ovary’. As such, it is of a great importance to the field. Among assisted reproduction techniques, molecular scaffolds are used to reproduce a three-dimensional ovarian environment to improve the efficiency of in vitro follicle culture techniques. The work presented here focuses on the comparison of the in vitro culture of human cortex pieces in a tube-shaped hydrogel made from chitosan and a basic system without this scaffold. In addition, the potential action of Activin A during the last part of this culture is studied.

Unfortunately, in current form it is difficult to follow.

The experiments are rather superficially analysed using limited amount of techniques: it is mainly based on follicle morphology and quality, follicular density (at primordial, primary and secondary stage) and diameter of only secondary follicles.

The number of ovarian cortical samples (5 ovaries from 5 patients?) and ovarian fragments per ovary (8 in culture and a control one) seems acceptable. Given the heterogeneity of the tissue in the same ovary, a second control fragment would be better.

But we have no information about the total counts of follicles in each stage at D0, D20 or D22 except a global count of 3519 (line 166) in the method paragraph or 3499 follicles in the discussion (line 419) and also nothing about the healthy tertiary follicles (number, proportion and diameters) at D20 or D22. Their presence is only specified in one sentence line 273 and no proof, like a photomicrograph in situ in the cultured cortex piece has been given.

A table has been added according to your recommendations (Table 1). It contains the number of follicles per category. This makes it easier to read the text. Histological cut photographs are in the supplementary file.

Furthermore, the status of the follicles in the cultured cortex was not tested by PCR: genes expressions would be interesting to analysed: Activin Receptor, AMH/AMHR, KIT/KITL, CYP19A1, CYP17A1 follistatin, FSH receptor, and activin bA and bB subunit … for example.

It’s a very interesting idea. We are not able to perform PCR analysis in this study because all tissues were fixed for histological and immunostaining analysis. We’re taking that into account for the next study.

AMH and progesterone secretion in the medium would be also interesting to measure.

Indeed, it would have been interesting.

Photomicrographs of the bioreactor in place and ovarian cortical tissue before and at the end of culture must be submitted with at least a picture of healthy antral follicle at the end of the culture. Likewise, pictures of the immunostaining are lacking.

Histological cut photographs are in the supplementary file.

The analyses that are presented in this report are not enough to make the final conclusions and there is no proof of oocyte functionality with at least meiotic competence after a maturation step for example. That makes me think that any of the protocols tested are actually producing functional oocytes. Nevertheless, this new device of cortex culture is always interesting and contribute to a better understanding of the in vitro folliculogenesis process. However questions remain: Given the complexity of making this bioreactor, is this protocol really an advantage? How transferable is the technology?

The bioreactor can be produced on a large scale. Moreover, it remains stable after conditioning for more than 2 years.

The introduction must be expanded by adding and discussing published data on

- Activin A and its receptor with their expression according to the stages and ovarian location.

“The use of Activin A combined with a proper culture system was reported to activate the primordial follicles and to initiate follicle growth. Activin A is a dimeric glycoprotein belonging to the transforming growth factor beta superfamily and its receptors are expressed in primordial, primary, and secondary follicles. It stimulates the granulosa cell proliferation, follicle, and oocyte growth and also accelerates oocyte maturation during culture of ovary cortex. »

- ovarian cortex culture in different species

We added this in the introduction: “in vitro systems have been developed to support the growth and maturation of follicles, a fortiori oocytes, from immature to final stage. The first live birth resulting from complete in vitro folliculogenesis was obtained in mice in 1996 by Eppig et al. and repeated in 2003. In large animals, the results were encouraging too; recently, the first pregnancy in goat from in vitro secondary follicle culture was reported. However, no team reported having obtained mature oocytes from primordial follicles. In human, metaphase II oocytes have been obtained, thanks to a multi-step culture system using Activin A, opening the possibility of in vitro fertilization as an option for women who cannot use the transplantation of their own ovarian tissue ».

Below are more details:

               Abstract

Line 21-22: precise that Activin A is provided only during the 6-8 last days of culture and not all along the culture.

Your remark has been considered. We have corrected the error in the manuscript.

Line 25 ‘significantly increased in all four groups’ is not exact (see results section).

This fits well with our results. Here are the two sentences of the result section:

  • “The proportion of primary follicles was significantly increased by culturing, p<0.001 (Figures 3-4)”
  • “The proportion of secondary follicles was significantly increased by culturing, p<0.001.”

Line 26 ‘Antral cavities were observed in the four culture groups’ no evidence in results.

The term “antral cavity” has been replaced by “tertiary follicle”. A more appropriate histological term.

Line 31 (p=0.006)

Your remark has been considered. We have corrected the error in the manuscript.

  1. Introduction

Line 45: ‘who cannot use the transplantation of their own ovarian tissue’

Your remark has been considered. We have corrected the error in the manuscript.

Line 46: The reference is McLaughlin 2018 and not 2010

We have checked the references and it is the 2010 article of this same team that we wish to cite here.

Describe first the most relevant published data about ovarian cortex culture then about individual isolated follicles culture systems while specifying the concerned species and distinguishing between review and research articles.

We added this in the introduction: “in vitro systems have been developed to support the growth and maturation of follicles, a fortiori oocytes, from immature to final stage. The first live birth resulting from complete in vitro folliculogenesis was obtained in mice in 1996 by Eppig et al. and repeated in 2003. In large animals, the results were encouraging too; recently, the first pregnancy in goat from in vitro secondary follicle culture was reported. However, no team reported having obtained mature oocytes from primordial follicles. In human, metaphase II oocytes have been obtained, thanks to a multi-step culture system using Activin A, opening the possibility of in vitro fertilization as an option for women who cannot use the transplantation of their own ovarian tissue ».

Line 47: to support in vitro folliculogenesis of isolated follicles

Your remark has been considered. We have corrected the error in the manuscript.

Line 55 : The use of Activin A combined with a proper culture system is reported to activate the primordial follicles and to initiate follicle growth during culture of ovary cortex + article reference

Your remark has been considered. We have corrected the error in the manuscript.

Line 59 : Bloise = Review / Telfer = individual follicle culture / Ding = cortex culture

Line 60 : It is a new paragraph: could you develop your concept of bioreactors with article references (perhaps for example Sacco et all, 2018 or other ones?)

We have completed the manuscript and added references to better understand the bioreactor.

Line 65 : with or without Activin A during the 6-8 last days of culture

Your remark has been considered. We have corrected the error in the manuscript.

Line 66 and 67 is a repeat of line 64-65. Choose!

Your remark has been considered. We have corrected the error in the manuscript.

  1. Methods

2.1. Ovarian tissue collection

Line 73: have you collected 5 ovaries from 5 patients?

Your remark has been considered. We have corrected the error in the manuscript.

Line 76: give the range of months

The range of months was 4-43.

“They all received a treatment by testosterone with a mean ± standard deviation duration of 25.3±18.4 (4-43) months”.

2.2. Bioreactor preparation

I do not know the technique of ‘3D chitosan scaffold’ for the support of ovarian follicles growth in cortex so I cannot judge this part of the manuscript. The reference articles indicated in the manuscript correspond to a patent and do not help my level of understanding of this technical approach. This tissue engineering approach has been already validated in 2008 by the same research team.

2.3. Tissue preparation and ovarian fragments culture

Line 109: for each ovary, One fresh fragment was taken and fixed in 4% paraformaldehyde at room temperature during 24h for histological evaluation (corresponding to the control at Day0)

In the schema: Fresh tissue assessment n=5       Tissue culture : 4 groups n=40

Lines 121-128 are the repetition of lines 116-119

Line 154: precise that Activin A was tested on individual follicles.

Thank you for pointing out these inconsistencies. We have taken all these remarks into account. We have modified the text accordingly. The changes are visible in the manuscript in "track changes".

2.5. Follicular density

Line 170: A Gougeon article reference?

This reference has been added

Line 177-178: explain the process of measure, rephrase

Indeed this paragraph is not clear. We have reworded it.

2.7. Follicular quality and morphology

Line 185: Precise if it is secondary or tertiary follicles

This concerns all follicles. To make it clearer we have created Table 1 reporting all follicles according to their stage and quality.

2.10. Statistical analyzes

Precise if you used SD or SEM in the data

We used the SD. “Continuous variables are described with the mean and standard-deviation and categorical variables with the count and percentage.”

  1. Results:

3.1. Follicular density

In freshly isolated uncultured tissue, the sum of the % of primordial, primary and secondary follicles is 89.5%. The remaining 10.5% correspond to the intermediate or tertiary follicles?

To make it clearer we have created Table 1 reporting all follicles according to their stage and quality.

Indicate in your text the Figure 3

The number of follicles counted for each stage is of great importance, precise it.

We have added Table 1 which summarizes this.

For group B the data are different between the text and the figure 3: Line 256: 44.5% in the text and ~40% in the figure 3. Line 263: 27.6% in the text and ~10% in the figure 3

We have completely redesigned the graphics. They are more relevant and correspond to the data.

Line 273: better placed in the 3.2 paragraph?

Line 275: remove intermediate?

Line 278-280: it is not the results but the description of the figure: Rephrase like ‘proportion of primordial, primary and secondary follicles in the control ovarian fragments (D0) versus ….’ given the heterogeneity of the tissue in the same ovary

We thank you for these remarks. We have taken them into account by modifying the manuscript.

It would be easier to write Group A (3D-/A-) Group B (3D-/A+) etc…

Indeed, it seems clearer. We have added it where possible in the figures of interest.

Line 279: ‘The proportion of primary follicles was significantly increased by culturing (B)’, it is not true for the primary follicles in groups A, B and C at D20 and group B at D22

The sentence was modified: “The proportion of primary follicles was significantly increased by culturing in group D at D20 and in groups A, B and D at D22 (Table 1 and Figure 3B).

3.2. Follicular growth

Line 283: ovarian cortex fragments in place of follicles

Lines 284-285: Specify the number of follicles

Line 283-285: Precise: No significant difference

Precise: secondary follicles

Thank you for pointing out these inconsistencies. We have taken all these remarks into account. We have modified the text accordingly. The changes are visible in the manuscript in "track changes".

3.3. Follicular quality and morphology

A table would be more readable

We have created the table according to your template.

Specify the follicular stages concerned. Is there differences in function of the follicular stages?

We have added Table 1 which summarizes this.

What is the observation in the control fragment?

The control fragment corresponds to the fresh tissue without culture (D0). The data are reported in Table 1 where the number of follicles counted per stage and their quality are listed.

Line 315-317: It is a legend and not a summary of the results

We have modified the text accordingly. The changes are visible in the manuscript in "track changes".

Specify in the legend that the ovarian fragments analysed are different in the D20 culture versus D22, although obtained from the same ovary.

We have modified the text accordingly. The changes are visible in the manuscript in "track changes".

3.5. Detection of Ki67 antigen

Line 325-327: if there is a difference between the group with 3D-structure and the group without, it would be interesting to distinguish these 2 groups in the analysis of the effect of Activin.

This is what has been done, here is the sentence from the manuscript: « in the group with Activin A compared to the group without (5 ± 17.1% vs 12.8 ± 22.5%, p = 0.108).”

The number of follicles counted in each condition must be specify.

Table 4 has been added. Slices were in the supplementary file.

3.6. Detection of Cleaved Caspase 3 antigen

Same remarks as for the previous paragraph

Table 5 has been added.

  1. Discussion

Line 347: specify the presence or not of activin during the last part of culture

Line 348: all follicular stages combined

We have taken all these remarks into account. We have modified the text accordingly.

Lines 354: there is also a mechanical and compressive action of the ovarian tissue.

The results must be better discussed and compared with the other culture of ovarian cortex.

It is difficult to discuss our results with other culture systems because current culture systems grow primordial follicles within the ovarian cortex so that they spontaneously activate and become primary follicles and then isolate the follicles. Our results are preliminary, which makes comparison difficult.

The impact of testosterone on the ovary must be more explain with the researches already published in animals (mice…) or sportive women.

We completed the discussion.

Lines 421-431 must be in conclusion

These sentences have been moved to the conclusion.

  1. Conclusion:

Too short and inexact

Line 433 replace beginning by end of the culture.

Add in the ovarian cortex fragments.

The conclusion has been modified according to your remarks. It is now more relevant.

Open your conclusion with the lines 421-431 and add sentences in order to think further on the use of this technology already successfully achieved in the mice (references in nature) but very far of the success in mammals and especially in humans

We had completed the conclusion: “Complete in vitro folliculogenesis was performed in mice and resulted in live birth [27, 28]. No birth has been described in mammals and human yet [29]. The results are promising but far from producing several good quality oocytes. Pregnancies and live births in mammals are essential to validate the current in vitro folliculogenesis techniques.

Reviewer 2 Report

The manuscript entitled « A new bioreactor to promote follicular growth with or without Activin

A » aims to investigate the effect of a new bioreactor and Activin A on human follicular

growth considering certain parameters (follicular growth, density, quality and morphology, a

histological study and an evaluation of Ki67 and Cleaved Caspase 3 immunoreactivity of

follicles, measurement of estradiol).

The content of the manuscript fits properly to the journal’s aims and scope.

The background of the study reflects a promising alternative for human female’s fertility

enhancement.

The introduction is well-written, stating the importance and the aim of the study.

Methodology and materials are clearly described.

The results are well interpreted and discussed, mentionning the study’s limits.

The language used is understandable and correct.

However, the manuscript contain several flaws, including :

Minor remarks :

- Authors should specify the name of the species used in the study for promoting

follicular growth (human follicular growth). Moreover, the title must include the sex of

the patients.

- The abstract should not include headings, and must include the background of the

study as well as the patients’ sex.

- Please, correct « Activing A » with « Activin A » in line 147 and 153.

- Insert a reference in line 170 and 396.

- Add a reference for the work mentionned in the paragraph from line 417 to 420.

- Please, make sure the abbreviations are well defined when first used in the

manuscript.

- The titles must be centered under the figures.

- Line 166 and line 419 : how many follicles were used in the study ? Please unify your

statement.

- The conclusion lacks information and statements.

- Please, make sure the term « gender » is used properly.

Major remarks :

- How does the work contribute in ovarian tissue cryconservation challenges ?

- The authors should imperatively take into consideration the patients’ testosterone

intake for evaluation, noting its effect on the study’s results.

- The samples’ lack and heterogenity are considered as a major weakness in the

present study.

- Does the study adhere to the rules of Helsinki’s Declaration guidelines ?

- It is advisable to present ethical committee approval as a supplementary file.

- Have the authors taken into consideration that the investigation involves vulnerable groups ?

- Have the authors taken into consideration SAGER guidelines (Sex and Gender equity in Research) ?

Author Response

Manuscript ID: reprodmed-2067408

Title: A new bioreactor to promote follicular growth with or without Activin A

Answers to Reviewer 2

The authors thank the reviewer for his/her attentive examination of the manuscript.

The manuscript entitled « A new bioreactor to promote follicular growth with or without Activin A » aims to investigate the effect of a new bioreactor and Activin A on human follicular growth considering certain parameters (follicular growth, density, quality and morphology, a histological study and an evaluation of Ki67 and Cleaved Caspase 3 immunoreactivity of follicles, measurement of estradiol).

The content of the manuscript fits properly to the journal’s aims and scope. The background of the study reflects a promising alternative for human female’s fertility enhancement. The introduction is well-written, stating the importance and the aim of the study. Methodology and materials are clearly described. The results are well interpreted and discussed, mentioning the study’s limits. The language used is understandable and correct. However, the manuscript contains several flaws, including:

Minor remarks:

- Authors should specify the name of the species used in the study for promoting follicular growth (human follicular growth).

Moreover, the title must include the sex of the patients.

- The abstract should not include headings, and must include the background of the study as well as the patients’ sex.

- Please, correct « Activing A » with « Activin A » in line 147 and 153.

- Insert a reference in line 170 and 396.

- Add a reference for the work mentioned in the paragraph from line 417 to 420.

- Please, make sure the abbreviations are well defined when first used in the manuscript.

- The titles must be centered under the figures.

- Line 166 and line 419: how many follicles were used in the study? Please unify your statement.

Thank you for pointing out these inconsistencies. We have taken all these remarks into account. We have modified the text accordingly. The changes are visible in the manuscript in "track changes".

- The conclusion lacks information and statements.

We don't understand your remark, could you clarify your idea please?

- Please, make sure the term « gender » is used properly.

Thank you for drawing our attention to this important point. We have changed the term in the manuscript.

Major remarks:

- How does the work contribute in ovarian tissue cryoconservation challenges?

Many patients cannot use their cryopreserved ovarian tissue because of the risk of reintroducing the initial pathology when their tissue is transplanted. We are in a therapeutic impasse. The idea would be to obtain in vitro oocytes from their cryopreserved ovarian tissue. This requires the establishment of in vitro folliculogenesis. This work contributes to the development of in vitro folliculogenesis techniques.

- The authors should imperatively take into consideration the patients’ testosterone intake for evaluation, noting its effect on the study’s results.

We completed the discussion: “Secondly, concerning our model, human ovaries were removed from patients who had testosterone before the surgery for a duration varying between four months to three years. This could have had an impact on the ovarian environment. Focusing on mice, it has been reported in the literature that testosterone treatment does not alter the quantity of follicle within the ovary and its effects are reversible [26, 27]. As for oocyte and follicle quality, the follicles were able to grow like follicles not exposed to testosterone and allow oocyte maturation. The oocytes resulted in the formation of cleaved embryos [28]. In a recent oral conference, E Telfer, obtained less follicles and a slower follicular growth in tissue exposed to male hormones than in non-exposed tissue (IVF worldwide, 2020).”

- The samples’ lack and heterogeneity are considered as a major weakness in the present study.

We agree with this remark, the heterogeneity is an important limit that is notified in the manuscript: “Moreover, another main limitation was the heterogeneity of the ovarian tissue concerning the ovarian reserve: a tissue with more follicles at baseline may be easier to activate than another tissue with a poor follicular reserve. “

- Does the study adhere to the rules of Helsinki’s Declaration guidelines?  It is advisable to present ethical committee approval as a supplementary file.

The ethical committee approval was presented in a supplementary file.

- Have the authors taken into consideration that the investigation involves vulnerable groups?

- Have the authors taken into consideration SAGER guidelines (Sex and Gender equity in Research)?

The project has been validated by the local ethics committee which considers the Declaration of Helsinki. The agreement of the ethics committee is in supplementary file.

Round 2

Reviewer 1 Report

My recommendation is to reconsider after revision because the results are not presented clearly enough for the moment. 

This manuscript concerns a work that seemed to me particularly interesting and well done (the count and analyse of follicles in cortex fragments is a hard work) and after this first correction the manuscript is now really improved and better written. Nevertheless, I think the manuscript might be released after some further improvements and I need more explanations especially in the results part.

Here are my remarks

ABSTRACT

Line 38: Add “Activin A did not seem to play a key role in in vitro folliculogenesis activation” in our culture conditions.

1.INTRODUCTION

Line 50:

You cannot write : ‘The first pregnancy in goat from in vitro secondary follicle culture was reported (De sa et al)’ :
It is in vitro culture of caprine early antral follicles (350µm) so at the beginning of the tertiary follicle, the antrum formation begin after 200µm in the goat.
They obtained a positive pregnancy diagnosis 30 days after embryo transfer. However, this animal was found to be no longer pregnant on day 45. So write, first beginning of gestation with abortion or not followed by a birth or first positive pregnancy diagnosis

2. METHODS

2.4. Histological evaluation

Line 164 + Line 197 + figure 2: did you used Fushia staining ? I don’t find explanation of the technique

3.RESULTS

3.1. Follicular density

Line 263:

The necessary number of follicles counted in each stage and condition have been well added, thanks to the table 1, as I requested. This table is mandatory but in order to make the manuscript more readable, this table could be added in the supplemental data if the authors prefer instead of in the results part. A summary sentence like “A total of 637 follicles in the fresh group and 1304 and 1578 follicles in the culture Days 20 and 22 respectively have been counted and classified in function of their quality (intact versus altered), their stage (…) and the culture conditions (with or without Activin, with or without 3D structure) (table S1)” could be added in the text of the manuscript for example.

Line 268, 275,282: For the term “by culturing”, please check English.

Line 264, 271, 278 : Please add “In freshly isolated uncultured tissue 62.4±27.35% of the follicles were identified as primordialwhatever their quality. Likewise, the proportion…

Line 284 : the sentence could be express with a global percentage or the range number of tertiary follicles in the fresh versus the cultured groups.

There is a mix between the legend of table 1 and figure 3 in the present form of the manuscript

Figure 3 is nice and make the results quite comprehensive. Just add the y-axis legend and the group D

Line 307: if you transfer the table 1 in the supplemental data, add in the legend of figure 3
Group A (3D-/A-), Group B (3D-/A+) etc…

Line 285-286: Do you compare group B+D versus A+C? Was the effect of activin A analysed in group B +D versus Fresh group?

Do you compare group C+D versus A+B? Was the effect of 3D culture analysed in-group C+D versus Fresh group?

Line 287-288: You specify a significance for the primary follicles, is this also valid for the secondary follicles? I think so.

3.2. Follicular growth

Line 315: Add “no significant difference of follicle growth or follicle diameter”

Line 315: Verify the count of 679 secondary follicles, perhaps 626, if I am not mistaken?

Please add “secondary follicles were measured” whatever their quality.

Line 318: Please add, secondary follicles

Line 319: 69.6 μm ±21.2

3.3. Follicular quality and morphology

With your table 2, the results are more readable. This table must be kept with the manuscript, but If you prefer, you can choose to use a figure with histograms (with D0 in reference) and put the table in supplemental data.

I am very surprised of the very low percentage of intact follicles in the cortex fragments, especially for primordial follicles. Is it useful or have you be very stringent or was it linked to the patient treatment perhaps? Please, give arguments. It seems to me that the percentage of viable follicles you found at D0 is significantly lower than in Ding's article for example. In the article of Ding et al. 2010, the proportion of viable follicles in the fresh uncultured tissue at day 0 is 92.9% and 71% of follicles in freshly isolated uncultured tissue were identified as primordial. Is it due to the testosterone treatment impact or your counting is perhaps more drastic ?

Some suggestions:

Table 2: “D20: 20 days of culture D22: 22 days of culture” are not necessary.

Table 2: Add “Fraction of intact follicles all stages combined counted in the cortex fragments ….

Table 2: you could replace standard deviation by SD in the table or specified in the legend that the data are expressed as mean +/- standard deviation.

Table 2:
If I use the data from Table 1, the number of follicles analysed is ok But I don’t find the same percentage of intact follicle: for example for group A D20, I find 35.4% instead of 42.7% etc…
Could you verify your data?

Please, indicate in the legend as a reference the fraction of intact follicles all stages combined in the fresh cortex fragment (D0)? Is it 11%? If the result is so, discuss in the text this percentage at D0 with a comparison with the cultured cortex fragments, which show a better quality of the follicles (reversal of the testosterone treatment impact)? (see above)

3.4. Estradiol secretion

The estradiol levels in each stage and condition have been well added, thanks to the table 3, as I requested. This mandatory table made the manuscript more readable however as the results are also well presented in the form of histograms and more comprehensive than in the first version under this presentation, this table could be added in the supplemental data if the authors prefer.

Table 3 : in the legend: culture media culture
specified in the legend that the data are expressed as mean +/- standard deviation.
if you transfer the table 3 in the supplemental data, add in the legend of figure 4
Group A (3D-/A-), Group B (3D-/A+) etc…

Line 351 : Significant increase in estradiol secretion

Line 353 and 358: I don’t understand your calculation of the estradiol secretion:
For example for the activin A supplemented groups,
what values do you add? Data at D6, D20, D22 are grouped?
146+2227+1738+680+854+1965=7610/6=1268?

Idem for the effect of 3D structure

3.5. Detection of Ki67 antigen

Line 378 replace within by between.

4. Discussion

Line 410 + 444: human ovaries ovarian cortex

Line 450: releasing of estradiol after cellular lysis from the cortex fragments cultured without structure?

Line 457: please replace “the same phenomenon” by the same proliferation and apoptosis pattern of the follicles (for example)

Author Response

Manuscript ID: reprodmed-2067408

Title: A new bioreactor to promote follicular growth with or without Activin A

Answers to Reviewer 1

Thank you for your pertinent review of the manuscript. Your review was thorough and relevant.

We thank you for the time you took to review our work. We hope you find the changes satisfactory.

We have answered the different questions in blue.

My recommendation is to reconsider after revision because the results are not presented clearly enough for the moment. 

This manuscript concerns a work that seemed to me particularly interesting and well done (the count and analyse of follicles in cortex fragments is a hard work) and after this first correction the manuscript is now really improved and better written. Nevertheless, I think the manuscript might be released after some further improvements and I need more explanations especially in the results part.

Here are my remarks

ABSTRACT

Line 38: Add “Activin A did not seem to play a key role in in vitro folliculogenesis activation” in our culture conditions.

We have completed the sentence of the manuscript.

1.INTRODUCTION

Line 50:

You cannot write: ‘The first pregnancy in goat from in vitro secondary follicle culture was reported (De sa et al)’ :
It is in vitro culture of caprine early antral follicles (350µm) so at the beginning of the tertiary follicle, the antrum formation begin after 200µm in the goat.
They obtained a positive pregnancy diagnosis 30 days after embryo transfer. However, this animal was found to be no longer pregnant on day 45. So write, first beginning of gestation with abortion or not followed by a birth or first positive pregnancy diagnosis

Modifications have been made.

  1. METHODS

2.4. Histological evaluation

Line 164 + Line 197 + figure 2: did you used Fushia staining ? I don’t find explanation of the technique

Fuchsia staining is a non-technical term that we use in our daily practice. It is a mistake on our part to have left it in the manuscript. It is actually a degeneration of the oocyte. The modifications in the manuscript and in the figure are made. Thank you for this remark.

3.RESULTS

3.1. Follicular density

Line 263:

The necessary number of follicles counted in each stage and condition have been well added, thanks to the table 1, as I requested. This table is mandatory but in order to make the manuscript more readable, this table could be added in the supplemental data if the authors prefer instead of in the results part. A summary sentence like “A total of 637 follicles in the fresh group and 1304 and 1578 follicles in the culture Days 20 and 22 respectively have been counted and classified in function of their quality (intact versus altered), their stage (…) and the culture conditions (with or without Activin, with or without 3D structure) (table S1)” could be added in the text of the manuscript for example.

We have lightened the text that is redundant with table 1 and left table 1 in the manuscript. We have also added your sentence in the introduction of this part of the results.

Line 268, 275,282: For the term “by culturing”, please check English.

Modifications have been made.

Line 264, 271, 278 : Please add “In freshly isolated uncultured tissue 62.4±27.35% of the follicles were identified as primordial” whatever their quality. Likewise, the proportion…

 Modifications have been made.

Line 284 : the sentence could be express with a global percentage or the range number of tertiary follicles in the fresh versus the cultured groups.

 Modifications have been made.

There is a mix between the legend of table 1 and figure 3 in the present form of the manuscript

 Modifications have been made.

Figure 3 is nice and make the results quite comprehensive. Just add the y-axis legend and the group D

 Modifications have been made. The legend of group D is below the figure and is in yellow on the histogram.

Line 307: if you transfer the table 1 in the supplemental data, add in the legend of figure 3
Group A (3D-/A-), Group B (3D-/A+) etc…

We prefer not to transfer the table to the supplemental data. We find that the information given within the manuscript is better for its comprehension. We thank you though for this proposal. 

Line 285-286: Do you compare group B+D versus A+C? Was the effect of activin A analysed in group B +D versus Fresh group?

We did compare groups B+D and A+C. However, we did not compare groups B+D with fresh. We have detailed this information in the manuscript.

Do you compare group C+D versus A+B? Was the effect of 3D culture analysed in-group C+D versus Fresh group?

We did compare groups C+D and A+B. However, we did not compare groups C+D with fresh. We have detailed this information in the manuscript.

Line 287-288: You specify a significance for the primary follicles, is this also valid for the secondary follicles? I think so.

The result is close to significance but not significant.

3.2. Follicular growth

Line 315: Add “no significant difference of follicle growth or follicle diameter”

Modifications have been made.

Line 315: Verify the count of 679 secondary follicles, perhaps 626, if I am not mistaken?

We thank you for identifying this major error. It is a copying error that we have corrected

Please add “secondary follicles were measured” whatever their quality.

Line 318: Please add, secondary follicles

Line 319: 69.6 μm ±21.2

Modifications have been made.

3.3. Follicular quality and morphology

With your table 2, the results are more readable. This table must be kept with the manuscript, but If you prefer, you can choose to use a figure with histograms (with D0 in reference) and put the table in supplemental data.

We thank you for your suggestion but we prefer to keep it in a table.

I am very surprised of the very low percentage of intact follicles in the cortex fragments, especially for primordial follicles. Is it useful or have you be very stringent or was it linked to the patient treatment perhaps? Please, give arguments. It seems to me that the percentage of viable follicles you found at D0 is significantly lower than in Ding's article for example. In the article of Ding et al. 2010, the proportion of viable follicles in the fresh uncultured tissue at day 0 is 92.9% and 71% of follicles in freshly isolated uncultured tissue were identified as primordial. Is it due to the testosterone treatment impact or your counting is perhaps more drastic ?

Indeed the difference with the literature is major. There are several explanations for this: the ovaries were not operated on by fertility preservation specialists, which can alter the ovary, the other important explanation is the fixative used. In Ding's article, the fixative is Bouin. It is much better at preserving the morphology of the follicles. We were not able to use it because it is forbidden in the lab.

Some suggestions:

Table 2: “D20: 20 days of culture D22: 22 days of culture” are not necessary.

Table 2: Add “Fraction of intact follicles all stages combined counted in the cortex fragments ….

Table 2: you could replace standard deviation by SD in the table or specified in the legend that the data are expressed as mean +/- standard deviation.

Modifications have been made.

Table 2:
If I use the data from Table 1, the number of follicles analysed is ok But I don’t find the same percentage of intact follicle: for example for group A D20, I find 35.4% instead of 42.7% etc…
Could you verify your data?

The statistician performed an aggregation of the data with a comparison of the aggregated data. This is an average of the averages. This is different from the general calculation made from Table 1.

Please, indicate in the legend as a reference the fraction of intact follicles all stages combined in the fresh cortex fragment (D0)? Is it 11%If the result is so, discuss in the text this percentage at D0 with a comparison with the cultured cortex fragments, which show a better quality of the follicles (reversal of the testosterone treatment impact)? (see above)

We have added an explanatory sentence in the methods to explain the absence of consideration of the fresh tissue (Line 199-200).

3.4. Estradiol secretion

The estradiol levels in each stage and condition have been well added, thanks to the table 3, as I requested. This mandatory table made the manuscript more readable however as the results are also well presented in the form of histograms and more comprehensive than in the first version under this presentation, this table could be added in the supplemental data if the authors prefer.

We prefer not to transfer the table to the supplemental data. We find that the information given within the manuscript is better for its comprehension. We thank you though for this proposal. 

Table 3 : in the legend: culture media culture
specified in the legend that the data are expressed as mean +/- standard deviation.
if you transfer the table 3 in the supplemental data, add in the legend of figure 4
Group A (3D-/A-), Group B (3D-/A+) etc…

Line 351 : Significant increase in estradiol secretion

Modifications have been made.

Line 353 and 358: I don’t understand your calculation of the estradiol secretion:
For example for the activin A supplemented groups, what values do you add? Data at D6, D20, D22 are grouped?
146+2227+1738+680+854+1965=7610/6=1268?

Idem for the effect of 3D structure

The statistician performed a multivariate analysis by grouping and comparing the groups with and without activin and the groups with and without structure.

3.5. Detection of Ki67 antigen

Line 378 replace within by between.

Modifications have been made.

  1. DISCUSSION

Line 410 + 444: human ovaries ovarian cortex

Modifications have been made.

Line 450: releasing of estradiol after cellular lysis from the cortex fragments cultured without structure?

Yes, we have made the modifications.

Line 457: please replace “the same phenomenon” by the same proliferation and apoptosis pattern of the follicles (for example)

Modifications have been made.

Reviewer 2 Report

The authors have addressed my comments and concerns in the revised manuscript, which has been greatly improved. I have no further comments.

Author Response

We thank you very much for the time you took to review our work and for your comments which improved our document.

Round 3

Reviewer 1 Report

The revised manuscript has been now greatly improved  and fit to the objectives of the journal. I accept it in the present form.

I have 2 remarks of text editing:

Line 435 human ovaries ovarian cortex
Line 485 : but